# Perceived Usefulness of Airborne LiDAR Technology in Road Design and Management: A Review

**Fazilah Hatta Antah [1], Muhamad Azry Khoiry [1,\*], Khairul Nizam Abdul Maulud [1,2] and Azlina Abdullah [3]**

1. Department of Civil Engineering, Faculty of Engineering and Built Environment, Universiti Kebangsaan Malaysia, Bangi 43600, Selangor, Malaysia; p106635@siswa.ukm.edu.my (F.H.A.); knam@ukm.edu.my (K.N.A.M.)
2. Earth Observation Centre, Institute of Climate Change, Universiti Kebangsaan Malaysia, Bangi 43600, Selangor, Malaysia
3. Centre for Research in Development, Social & Environmental (SEEDS), Faculty of Social Sciences and Humanities, Universiti Kebangsaan Malaysia, Bangi 43600, Selangor, Malaysia; azlina_ab@ukm.edu.my
* Correspondence: azrykhoiry@ukm.edu.my

**Abstract:** Airborne light detection and ranging (LiDAR) surveying technology plays an important role in road design, and it is increasingly implemented in the design stage. The ability of LiDAR as a remote sensing technology to be used in non-accessible places (i.e., hilly terrain, steep slope) makes it a powerful tool, and it has the potential to provide benefits that simplify existing design processes for designers and practitioners. This paper reviews the application of airborne LiDAR in road design and factors including items from the perceived usefulness of technology. The context of the future direction of LiDAR technology is highlighted in civil engineering road design, roadway inspection and as-built documentation. The implementation of this technology is expected to assist the end-users in developing more manageable planning for road construction and thus to ensure the usage of LiDAR technology is enhanced from time to time, especially in Malaysia.

**Keywords:** airborne LiDAR; road planning; sustainability; road design; road management

## 1. Introduction

There are many issues faced by transportation agencies. Among these are the difficulty in determining the route for road alignment according to geometry and road design standards in different types of terrains [1,2]; the limitations of existing surveying measurement methods in providing accurate [3] and reliable information [4], especially in large areas and under thick canopies [5]; the inability to estimate the amount of cutting and filling in road earthworks [6] in order to select a road design that can provide optimal costs at the road planning stage [7,8]; the inability to produce a road design that can minimize the environmental impact, particularly in terms of natural river disruption [9], tree felling [10], wildlife settlement and the influence of landscapes on inhabitants' dwellings [11]; the inability to notice the risks and problems of an undulating terrain [12]; the inability to detect any slope failures [13] and the inability to provide access to other users in the road design when the topographic data are missing [14].

Most road design works are performed based on topographic data obtained through traditional methods such as using Total Station (TS) surveying measuring equipment. However, there are several drawbacks in using TS, particularly in locations where the landscape is undulating or mountainous. Aside from that, the distance between the location spot heights captured using TS is inconsistent [15]. Due to time constraints, only a limited number of points can be measured [3] in this survey. When topographical differences between survey stations are disregarded, the design of curved roadways can be challenging in terms of the steepness of the horizontal profile, and the incorrect estimation of the earthwork volume may occur [16,17] at the same time. Apart from that, there is also

incomplete measurement data information due to the constraints encountered during the field survey [18].

There are also a great deal of missing spot height data, particularly in areas which workers with TS have failed to enter or observe [19,20]. Furthermore, if inexperienced surveyors undertake the surveys, the acquired data are vulnerable to human error [21], such as the instruments being in poor condition, obstacles from steep slopes [22] and a lack of entrance access to the field measurement site [20]. Topographic features are primarily non-automated and require many individuals to check and observe the data collection process [23]. This paper tries to describe the usage of light detection and ranging (LiDAR) and its advantages in road project management, which encompasses the planning, design and maintenance phases.

Airborne LiDAR has become an instrument for survey purposes and is used to measure and collect unlimited and dense topographic data at a very high speed [24]; furthermore, as a tool in remote sensing, it is used in observation devices that provide information about the surface of the earth and the surface of the water [25]. The LiDAR system consists of a laser scanner; a range unit (ranging); control, monitoring and recording devices; a differential global positioning system (DGPS) and orientation position systems, namely inertial navigation systems (INS). Laser scanners play a role in determining physical attributes such as altitude, and the DGPS and INS can record the location and position of the target object through aircraft (x, y, z) three-dimensional geological coordinates based on the World Geodetic System of 1984 (WGS84) data, as presented in Figure 1 [26–28].

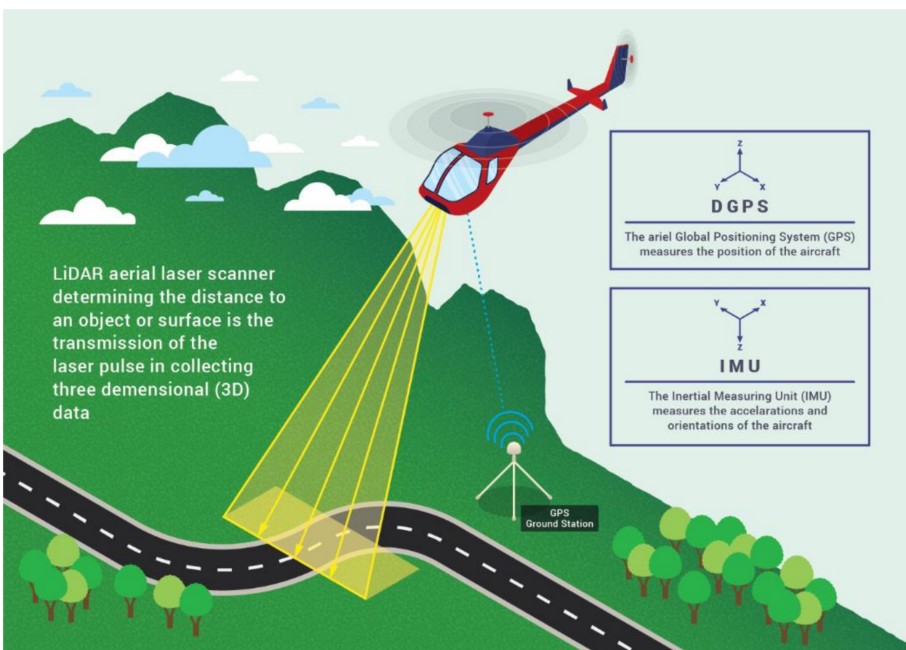

**Figure 1.** Typical airborne LiDAR system.

Figure 1 shows the LiDAR airborne system consisting of the three main components of the scanner laser, DGPS and Inertial Measurement Unit (IMU). The LiDAR sensor generates short-duration laser pulses, transmits them to the ground, scans the ground beneath while firing pulses, receives the return signal and measures the return pulse's time of travel. With the advent of the Global Location System, the precise positioning of aerial lasers has become achievable (GPS). The IMU sensor monitors the aircraft's accelerations and orientations, which are significantly higher than the Global Navigation Satellite System (GNSS) epoch (for example, 400 Hz) [24].

The following is a breakdown of the paper's structure: Section 2 outlines the research approach; Section 3 compares between TS, LiDAR and an Unmanned Aerial Vehicle (UAV); Section 4 describes the implementation of LiDAR in road design based on previous studies;

Section 5 describes items that explain the factors and items of the perceived usefulness of this technology; Section 6 discusses the future direction of LiDAR technology in civil engineering road design; Section 7 details LiDAR technology for roadway inspection and as-built documentation; Section 8 elaborates on the limitations; and Section 9 presents the work's conclusion.

## 2. Methodology

In research, a literature review is a critical component. This literature review analyzes and synthesizes data from prior studies to improve our knowledge and methodically describes the novelties of the research to improve our understanding of that knowledge [29]. This review paper is based on a comprehensive evaluation of the literature from electronic sources such as journals, books, publications from international organisations, and articles. The articles were located in the databases of Web of Science, Google Scholar, and the e-Journal Portal of Universiti Kebangsaan Malaysia (UKM). A total of five topics were investigated as a result of the review. After reviewing, five topics were identified; (1) the comparison of LiDAR airborne technology to other surveying technologies; (2) studies that were conducted previously in the context of road project management involving road design using LiDAR airborne technology; (3) variable items such as information quality, user satisfaction, improvement of job performance, system quality, and top management support that can be used to study the acceptance of airborne LiDAR technology among road designers; (4) the direction of using airborne LiDAR in civil engineering road design and the advantages of LiDAR advantages in road designs in sloped and hilly areas, in locating drainage structures, in monitoring road connectivity, providing topographic data for road planning and design, and minimizing a road's environmental impact; and (5) the use of airborne LiDAR technology in roadway inspection and as-built documentation, which includes road asset inventory, road modeling, road geometry assessment, and road inspection. Figure 2 shows the overall methodology used in this study.

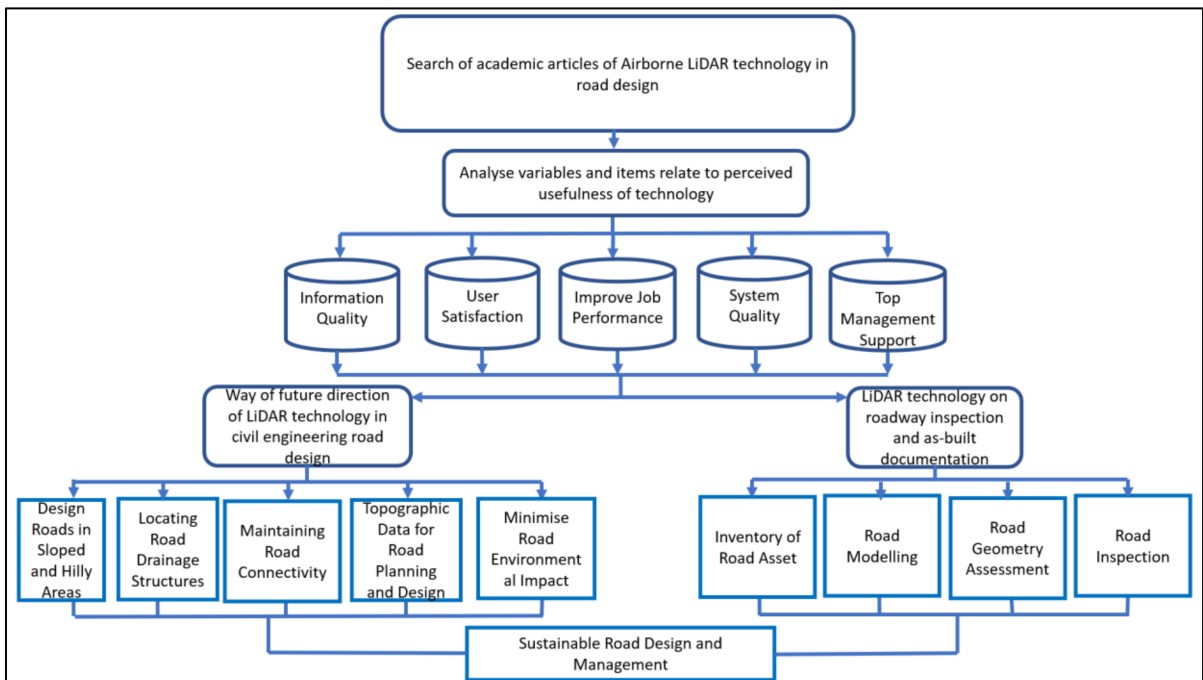

**Figure 2.** Overall methodology.

### 3. Comparison between Total Station (TS), Airborne LiDAR, and Unmanned Aerial Vehicle (UAV)

Many surveying measurement methods can be used in road design applications. Among the survey methods used to aid transportation agencies and road designers are TS, Airborne LiDAR, and UAV photogrammetry. However, there are weaknesses in TS that were identified according to Myers et al. [15]; for example, using TS in stream bank areas lead to difficulties in measurement due to overhanging or undercut banks. Although TS is able to measure a large area according to S.I. Mohammed [30], it prolongs the measurement time of collecting the data according to V.Č. Aksamitauskas et al. [31] and thus increases manpower, increasing the measurement cost according to Zulkipli et al. [32]; however, T.B. Afeni and F.T. Cawood [33] described TS as a costly, high-precision measurement tool that is able to measure with an error up to 1 mm over a hundred meters.

According to C.H. Grohmann et al. [32], one of the advantages of the airborne approach is that it is able to save data acquisition time compared to the field survey measurement that generates Digital Elevation Model (DEMs) data automatically with very high accuracy. The capability of airborne LiDAR is highly evident and significant in measuring fields over a wide area and provides a broader spatial extent, according to S. Landry et al. and C.H. Grohmann et al. [34,35], and it is able to record the data of a surface with a 1 m measurement accuracy and of producing a point density within 2–20 points per square metre and a vertical accuracy assessment of $\pm 20$ cm, according to J. Muir et al. The laser capability of airborne LiDAR which penetrates the canopy indeed helps to give accurate readings on the terrain, according to J.R. Roussel et al. [3,36]. However, behind the high accuracy of the data, drawbacks are identified where the observations can be affected during cloudy and raining days according to G.J.P. Schumann et al. [37], and the approach involves a high cost according to Grohmann et al. [35].

Surveying using the UAV method, according to M. Khanal et al. and K.L.A. El-Ashmawy [19,20], has a lower measurement cost because of the smaller number of workers required compared to the number of workers required by TS measurement. According to M. Khanal [19], UAV saves measurement time in the field especially in terms of the DEM data generated automatically by UAVs. However, according to M. Khanal et al. [19] and Pellicani et al. [3,19], there are drawbacks in terms of the constraints of data observations that can only be measured in a small measurement field with a vantage point that can affect the accuracy of the UAV readings. In contrast, according to K.L.A. El-Ashmawy [20], the capabilities and obstacles during data observation by UAV can be affected by cloudy and rainy conditions.

After comparing the three measurement tools above, the airborne LiDAR surveying measurement method was chosen for data observation for the design of new roads in unexplored areas. New road design requires a wide data observation field to see the connectivity between the new road and the existing road. Therefore, airborne LiDAR is seen to be able to speed up the road planning and design process because the data are generated in a shorter time, especially in rolling and mountainous terrain areas where a very high elevation accuracy can be achieved. Airborne LIDAR can overcome the dependence on workers to prepare data for long and wide road design, especially highway construction. The comparison between TS, airborne LiDAR and UAV photogrammetry is shown in Table 1.

**Table 1.** Comparison among Total Station (TS), Airborne LiDAR and UAV photogrammetry.

| Aspect\Method | Total Station (TS) | Airborne LiDAR | UAV Photogrammetry |
|---|---|---|---|
| Restriction on data collection | Problematic due to overhanging and undercut measurements | Limitation under cloudy and raining | Limitation under cloudy and raining |
| Period of surveying data collection | Time-consuming | Conserves time | Conserves time |
| Coverage area | Covers a wide area | Covers a wide area | Covers a small area |
| Accuracy of data | High-precision measurement with an error up to 1 mm over a hundred meters | 1-m measurement accuracy of surface | Constraint by photographic image |
| Cost | High labour cost and equipment in wide area | High cost | Low cost |

## 4. Implementation of LiDAR in Road Design Based on Previous Studies

R.A White et al. [38] proposed 1 m Digital Elevation Model (DEM) data and compared it with the a road's centre line using a field survey that provided up to 1.5 m accuracy. White et al. [38] study shows that LiDAR is suitable and accurate to assess road position, gradient, and road length and thus can be used as a road inventory for existing roads and road mapping.

M. Contreras et al. [17] developed a computerized model using LiDAR DEMs to obtain the quantity of cutting and filling volume values in earthworks. This study focuses on the design of roads that comprise hilly and rugged terrain. The calculation of earthworks in this road design is taken at an estimated distance of 1 m as required for the value of the conducted earthworks.

M. Saito et al. [13] used an analysis of DTMs with a grid measurement of 10 m and analysed these automatically on the computer. The road design produced and proposed a slope analysis. Spline interpolation was used to balance the value of the cutting and filling quantity in the earthwork. This study avoids the design of roads being built in locations that are at risk of landslides.

The study conducted by Z. Azizi et al. [39] used the Support Vector Machine (SVM) road extraction algorithm model with integrated LiDAR DEMs to classify road and no-road areas. The results of this study help to update accurate and up-to-date information on the road inventory to assist in road design management.

A. Ekay et al. [40] developed a 3D computerized road program to optimize the cost of earthworks for each possible road alignment using a linear programming approach to LiDAR DEMs. A. Ekay et al. [40] estimated the average annual amount of silt sent into the stream through the road network. Extraction from LiDAR was also used by A. Ekay et al. [40] to create current road mapping.

Parsakhoo and M. Jajouzadeh [41] used Dijkstra's algorithm with LiDAR DEMs to optimize road alignment and therefore estimate construction costs. This study determined the best route and road alignment.

B. Matinnia et al. [16] used the triangular irregular networks (TIN) algorithm to generate geometric alignments and road cross-sections using LiDAR DEMs in Autocad Civil 3D road design software. This study shows that the geometric parameters of horizontal and vertical roads can be generated under dense canopy without conducting a field survey.

B. Matinnia et al. [4] used the TIN technique to create road cross-sections using Autocad Civil 3D road design software. Then, the total quantity of cutting and filling was determined using the earthwork volume formula. The results of this study can help in determining a road's alignment, the availability of road fill cut, and the fill amount when making earthworks for road design, which is helpful for terrain analysis and road mapping.

S. Buján [12] used a hybrid classification approach integrating LiDAR DEMs to distinguish between paved and dirt roads, including bare land and low vegetation. The road was separated into two types—paved roads and unpaved roads—which is critical for road network mapping.

From previous studies, it can be concluded that airborne LiDAR is very important in the road design stage. Accurate information obtained from LiDAR [42] has assisted in terms of road position, road gradient, road length, ground slope, terrain roughness, shape extraction, determining paved roads and dirt roads, vertical and horizontal profiles, and cross-sections. It is also essential in determining road mapping, soil quantity calculation in earthworks, road geometry, road networking, and road alignment routes. Table 2 below summarizes the implementation of LiDAR in road design based on previous studies.

**Table 2.** Implementation of LiDAR in road design based on previous studies.

| Author(s)/Year | Country | LiDAR Data | Methodology | Parameter | Findings |
|---|---|---|---|---|---|
| R.A. White et al., 2010 | California, United States | DEM | • The road's LiDAR was manually digitized in ArcMap. | • Road position. • Road gradient. • Road length. | • LiDAR DEMs compares the centre line of the road. • Data collected as a road inventory record. • LiDAR can be used to map the existing road network. |
| M. Contreras et al., 2012 | Idaho, United States | DEM | • Model created with computerized software. • Automatic software earthworks are estimated by automatic software at a 1 metre spacing. | • Ground slope. • Terrain roughness. | • Quantity in volume number of earthworks. |
| M. Saito et al., 2013 | Funyu, Japan | DTM | • Computerized automatic road design by analysing 10 m grid DTMs. • Slope analysis. | • Quantity of cut and fill. | • Reduce earthworks. • Avoids road alignment in landslide-prone locations. |
| Z. Azizi et al., 2014 | Golestan, Iran | DEM | • SVM algorithm to extract road. • Image segmentation. | • Shape extraction. | • Road mapping of existing road. |
| A.Ekay et al., 2014 | Oregon, United States | DEM | • 3D road alignment model. | • Quantity of cut and fill. | • Quantity in volume number of earthworks. • Road network. |
| Parsakhoo and M. Jajouzadeh 2016 | Golestan, Iran | DEM | • Dijkstra's algorithm to optimize road path. | • Road length. • Road gradient. • Road construction cost. | • Determines route for road alignment. |
| B. Matinnia et al., 2017 | Bahramnia, Iran | DEM | • Surface created in road design software Autocad Civil 3D by triangulated irregular network (TIN). | • Vertical profile. • Horizontal profile. • Cross-section. | • Determines road geometric properties under dense canopy. |
| B. Matinnia et al., 2018 | Bahramnia, Iran | DEM | • Surface created in road design software Autocad Civil 3D by triangulated irregular network (TIN). • Earthwork quantity estimation. • Estimation of cost and time by LiDAR. | • Vertical profile. • Horizontal profile. • Cross-section. • Quantity of cut and fill. • Time and cost. | • Road mapping. • Quantitative terrain analysis. |
| S. Buján et al., 2021 | Trabada, Spain | DTM | • Breiman's RF algorithm to normalize elevation. • HyClass (hybrid classification) of road. | • Paved road. • Dirt road- | • Road network. |

## 5. Perception towards of Factors and Items Regarding the Usefulness of Technology

The selected factors regarding the use of technology based on the previous studies can be sorted into five main categories: information quality, user satisfaction, improved job performance, system quality, and top management support. The items contained in these factors are briefly described in Table 3.

### 5.1. Quality Information of Technology

Based on the factors of information technology in the study conducted by Song et al. [43], it is claimed that the existing use of technologies would help to provide information to calculate components such as the quantity of the design work produced, whereas Diop et al. [44] also confirmed that technology aided in the collection and dissemination of user information in an efficient system.

**Table 3.** Factors and items regarding perceived usefulness of technology based on previous studies.

| Factors | Items | Author(s) | Year(s) |
|---|---|---|---|
| Information quality | Good quality of information results in better selection. | Song et al.<br>Diop et al. | 2017<br>2019 |
| | Information is presented in a useful format. | Song et al.<br>Wang and Chen | 2017<br>2018 |
| | Data information content is valuable and meets our needs. | Song et al.<br>Juan et al.<br>Lozada Martnez et al. | 2017<br>2018<br>2019 |
| | Essential for learning and understanding content. | Juan et al.<br>Altabe Lozada Martnez et al. | 2018<br>2019<br>2019 |
| | Allows complex problems to be solved. | Ciolkowski et al.<br>Altabe<br>Mezhuyev et al. | 2008<br>2019<br>2019 |
| User satisfaction | Satisfied with the technology. | Mohaghehgi et al.<br>Song et al. | 2013<br>2017 |
| | Enables users to visualize and adapt with the real environment. | Juan et al.<br>Madlala et al. | 2018<br>2020 |
| | Saves time. | Mezhuyev et al.<br>Tan and Hsu | 2019<br>2018 |
| | The features of new technology releases. | Hong et al. | 2011 |
| Job performance | Usage of technology will be advantageous to work. | Zerei et al.<br>Man et al.<br>Mezhuyev et al.<br>Tan and Hsu | 2017<br>2020<br>2019<br>2018 |
| | Useful for completing a task. | Khamaruddin et al.<br>Man et al.<br>Zettel | 2017<br>2020<br>2005 |
| | Tool in the work process. | Chnyamurindi and Low<br>Wallace and Sheetz | 2010<br>2014 |
| | This measure has been tested and validated. | Wallace and Sheetz | 2014 |
| | Enhances effectiveness at work. | Son et al.<br>Turnet et al.<br>Caron Fasan et al.<br>Zerei et al.<br>Horton et al. | 2015<br>2010<br>2020<br>2017<br>2001 |
| | Technology facilitates learning. | Lozada Martnez et al.<br>Kumar et al. | 2019<br>2020 |
| | Increases my interactions with others. | Kumar et al.<br>Nejkovic and Tosic | 2020<br>2018 |

**Table 3.** *Cont.*

| Factors | Items | Author(s) | Year(s) |
|---|---|---|---|
| System quality | Increases productivity. | Altalbe<br>Mezhuyev | 2019<br>2019 |
| | The system is functional and compatible. | Iivari<br>Fedhel et al.<br>Rodger<br>Tobarra et al.<br>Dart et al. | 1995<br>2019<br>2020<br>2020<br>2020 |
| | Assist with meta-modelling in software process development. | Mezhuyev et al. | 2018 |
| Top management support | Top management should provide training. | Iivari<br>Song et al. | 1995<br>2017 |
| | Vendor should be compatible. | Iivari | 1995 |
| | Top management should provide human resource support for technology. | Song et al. | 2017 |
| | Top management should provide consultation. | Song et al.<br>Fadhel et al.<br>De Oca et al. | 2017<br>2019<br>2014 |
| | Top management should provide financial support. | Song et al. | 2017 |

In the study of Song et al. [43], information is presented in a good format using the application of software technology. Their approach can document all project information for everyone's use. In the study of Wang and Chen [45], their approach incorporates product feature information such as brand and technical engineering features.

Data information content is valuable and meets the project needs of users, as proven in the studies of Song et al. [43], Juan et al. [46], and Lozada-Martínez et al. [47] into making work content more interesting and suitable for learning.

Technological items are essential for learning and understanding content. Juan et al. [46] stated that digital images produced by computer technology in real time and in three dimensions would improve understanding. According to Altalbe [48], virtual simulation can increase understanding in a similar manner to seeing in real life. Furthermore, Lozada- Martinez et al. [47] mentioned that technology can be used more effectively than traditional methods.

According to Ciolkowski et al. [49], technology allows complex problems to be solved with visual colorization through the help of technology, allowing researchers to overcome the difficulty of analysing data. According to Altalbe [48], technology can assist in completing complex experimental tests virtually without using real experimental tools. Mezhuyev et al. [50] used Search-Based Software Engineering (SBSE) to solve the problem of high-complexity mathematical optimization.

*5.2. Users Satisfaction with Technology*

According to previous research by Mohagheghi et al. [51], users are satisfied with technology if the performance tools of the technology on large-scale projects are proven to work excellently, and according to Song et al. [43], organizational support is essential in determining users' satisfaction with technology.

According to Juan et al. [46], technology provides an alternative for users to visualize and experience the real work environment by improving the in-depth examination of the construction interface. According to Madlala et al. [52], an interactive application on a phone helps to visualize the subject to be taught in online learning.

Users believe that the use of technology will reduce the time needed to complete certain tasks. According to Tan and Hsu [53] and Zarei [54] et al., technology speeds up the execution of work by reducing the time it takes to complete it. Mezhuyev et al. [50] mentioned that the use of technology was able to reduce the length of the design process carried out in their study.

Finally, Hong et al. [55] mentioned that programming is developed to produce futuristic features because users prefer features that are easy to use.

### 5.3. Technology Improves Job Performance

According to Man et al. [56], technology improves security and reduces human error. According to Mezhuyev et al. [50], comprehensive achievements can be achieved with technological efficiency by enhancing the development of the software used. Tan and Hsu [53] explained that technology has an advantage in terms of improving the knowledge and skills of employees.

Khamaruddin et al. [57] found that a technological platform is great for monitoring individual work progress to ensure each task is completed. Man et al. [56] and Zettel [58] claimed that technology helps in completing high-quality work.

According to Chinyamurindi and Louw [59], technology is an effective tool to help employees to understand the work process with the production of images. According to Wallace and Sheetz [60], technology is a tool for measuring the process and progress of work, and it can also measure cost.

Turner et al. [61] explained that technology improves work practices by referring to the completion time of work and quality of work. Caron-Fasan et al. [62] stated that technology-assisted work practices can streamline the work process by improving the understanding of the work. According to Son et al. [63], technology facilitates the design process, allowing it to become more organized. Horton et al. [64] found that technology accelerates the progress of work, making the process more effective.

According to Zarei et al. [54], technology facilitates the learning process and encourages interaction between learners that will benefit their work. According to Lozada-Martnez et al. [47], communication and interaction occur in an organization without the need to meet, and this can be done with self-training due to the efficiency of technology. Kumar et al. [65] and Nejkovic and Tosic [66] found that users agreed with the point that technology is able to facilitate learning regardless of time and space and it is easily accessible.

### 5.4. Quality of Technology System

According to Iivari [67], computer-aided software for technology helps to enhance the functionality of the system and its compatibility. Fadhel et al. [68] found that their developed system met the needs of users and further facilitated use. According to Rodger [69], a developed system can maintain the safety of the user, which can reduce the accidents caused by the inefficiency of the user. Tobarra et al. [70] found that a system with virtualization technology can increase the efficiency of a user, allowing them to adapt to each changing environment. According to Dart et al. [71] and Mezhuyev et al. [72], a system should provide features that enable more intentional designs which can be accepted and used by users. Mezhuyev et al. [72] explained that a system can benefit from metamodelling by recognizing the explicit modelling language and indicating how the model should be developed with features and capabilities.

### 5.5. Top Management Support for Technology

According to Son et al. [63], training should be provided to employees so that it is more available in the event of difficulties and, at the same time, to increase user confidence in the use of the system effectively. According to Iivari [67], training should cover in-house training, self-studies, and vendor courses.

Iivari [67] recommended that vendors appointed by the management for the implementation of technology in the workplace must be compatible. According to Song et al. [43], adequate human resource support is needed to provide trained staff with a system that is able to support other staff. Moreover, according to Song et al. [43] and Fadhel al. [63], consulting should be provided by top management to provide direction to settle any misunderstandings and—more importantly—for the smooth running of the system. According to De Oca et al. [73], guidelines are needed to ensure the better understanding of the users towards the technology used, which indirectly facilitates the purpose of learning and teaching and ensures the quality of the system. Lastly, Song et al. [43] also stated that top management must allocate sufficient financial funds for the procurement and use of technology.

### 5.6. Relation among Factors and Items in Implementing Airborne LiDAR Technology for Road Design and Management

The previous studies mentioned above are related to the user acceptance of the use of technology including information quality, user satisfaction, job performance improvement, system quality, and top management support. However, to look into the acceptance of airborne LiDAR technology by its users, a complete procedure or guideline needs to be created. This should serve the purpose of understanding the shape or design of a road that uses this LiDAR technology.

As we know, LiDAR is suitable and practical for large areas, especially those with mountainous terrain. The items discussed above assisted in constructing a questionnaire in terms of training needs and road design management modules. Further studies need to be conducted, mainly with a large-scale transportation agency for road designs. LiDAR technology has been seen to help agencies in the planning stage to determine road networking for a road design. A transportation agency can therefore study the form of training needed by staff, especially for those involved in the planning for optimizing the application of airborne LiDAR. The guidelines and training module are beneficial in developing knowledge among engineers, helping to strengthen airborne LiDAR when used for design management in the government and private sectors.

## 6. Future Direction of Airborne LiDAR Technology in Civil Engineering Road Design

Airborne LiDAR has a great deal of potential, especially in road design, thanks to outstanding civil engineering research. Transportation agencies will benefit from this advantage, as stated in full below.

### 6.1. Minimize Road Environmental Impact

In terms of road design and management, airborne LiDAR can also be used as a spatial model to identify the form of soil erosion because of the risk of soil erosion posed by infrastructure construction projects [74,75].

Part of the scope of environmental monitoring is to monitor sedimentation at the project implementation stage [76,77]. Airborne LiDAR can minimize road environment issues, as stated in a study by A.E. Akay et al. [40], by using an equation to calculate treated sediment and cut slope sediment; for example, the 3D Forest Road Alignment Optimization calculates the quantity of sediment delivered to a stream from road networks. In the work by Contreras et al. [17], a computerized model was developed by taking several factors into consideration, such as ground slopes and differences in terrain ruggedness conditions, to reduce the volume of cutting at the slope.

J.M. Vilbig et al. [78] found that the quasi-network flow model, by considering side-slopes and physical blocks in the terrain, can assist in road design by reducing the volume of cutting at the slope. W. Chen et al. [79] stated that LiDAR data and slope images have assisted in identifying a landslide-prone region. Meanwhile, a study by T. Görüm [80] showed the effectiveness of the Landslide Inventory from LiDAR to reduce landslides at a road work area. Lastly, M. Saito et al. [13] explained the Landslide Risk Map, which was developed to detect potential shallow slides near to the road.

### 6.2. Designing Roads on Slopse and in Hilly Areas

Airborne LiDAR is valuable for road design on slopes and in hilly areas. In their study, P. Jagodnik et al. [81] discovered that the type and shape of soil geological engineering properties can be integrated to detect gravitational mass movement and rockfall occurrence from mapping using a visual interpretation from LiDAR. In comparison, R. Pellicani et al. [82] and D. Gadone et al. [83] identified characteristics of soil failure signals and hence detected soil instability by mapping from LiDAR. The results from C. Chen et al. [84] also showed that the progressive TIN Densification (PTD) filtering used on LIDAR data is suitable to be used on hilltops and steep slopes to detect ground movement conditions.

### 6.3. Locating Road Drainage Structures

It is challenging and difficult to define a drainage system (i.e., drains and culverts) for road and infrastructure design [85]. However, a study conducted by J.B Lindsay and K. Dhun [86] mentions that the use of the Least-Cost Breaching method on airborne LiDAR data actually helps in determining surface drainage channel patterns, particularly those found under bridges and culverts.

In addition, R. Li et al. [87] showed that drainage patterns could be determined using LiDAR-derived hydrologic DEMs by using a geospatial method. In comparison, J. Roelans et al. [88] developed LiDAR Dropout Modelling by separating LiDAR data into ditch points and non-ditch points, then putting ditch points together to form a 2D polygon object to determine ditch drainage. Another study was also conducted by J.B Lindsay et al. [89], who introduced a feature-preserving smoothing method for drainage pattern flow to store the information of drainage features on LiDAR data.

### 6.4. Maintaining Road Connectivity

When developing new road plans and designs, it is essential to maintain road connectivity to ensure a safe and sustainable traffic network. Several research studies show the effectiveness of airborne LiDAR in aiding in this road connectivity. The study by M. Sharma et al. [26] showed that LiDAR topographical data from DEM and BEM are valuable for creating road network infrastructure projects since they are highly accurate. Meanwhile, S. Buján et al. [12] found that road-mapping between paved and unpaved roads from DTM aids in road connection and new road design as it transits from new roads to principal roads. Meanwhile, L. Barazzetti et al. [90] used DTM to obtain road centreline information and elevation readings to model a new road network connecting existing and new roads. K. Karila et al. [91] also compared this to aerial imagery, where LiDAR-based road-mapping delivers exceptionally high accuracy in road network monitoring.

### 6.5. LiDAR Data for Road Planning and Design

In the development of road designs in wooded, thick-canopied, and mountainous areas, airborne LiDAR can provide very intensive terrain data. Studies conducted by B. Matinnia et al. [16] found that the remarkable accuracy of LiDAR data aids road design by representing the vertical and horizontal alignment of the road without time-consuming field surveying methods in broad areas. Meanwhile, B.P Reis et al. [92] showed that LiDAR can produce main images in places with dense tree cover to sustain forest regeneration followed by road network design to avoid this area in order to save the environment.

A study was also conducted by B. Bigdeli et al. [93] that used Invasive Weed Optimization presented for LiDAR point interpolation to improve accuracy when creating a DTM at the forested canopy and steep terrain. In addition, the study by N. Fareed and C. Wang [94], which incorporates Drainage Structure Mapping Algorithm, proposed that culvert modified DEMs could be obtained for hydrological studies on highways. Meanwhile, R. Pu and S. Landry [95] used multi-seasonal Pléiades satellite imagery and airborne LiDAR data to provide accurate tree mapping to maintain a sustainable development area.

*6.6. Overall Future Direction of Airborne LiDAR Technology Data for Road Planning and Design*

Overall, there are many advantages of road design in management aspects, as reviewed in the literature by other scholars. However, its future development needs to be studied and reviewed by considering the geographical factors of Malaysia, as Malaysia is a tropical country that experiences heavy rainfall throughout the year [96]. Monsoon changes have caused floods in the nation every year [97]. As such, an efficient road transportation system is a requirement in low-landed areas, in which flood is likely to occur. At the same time, roads in mountainous areas and on steep slopes may possibly collapse during heavy rains [98]. Therefore, airborne LiDAR technology becomes an essential tool to predict and control road safety in such areas. Generally, several issues regarding the usage of LiDAR in government agencies such as the Public Works Department of Malaysia need to be reviewed. For the maintenance department [99,100], the use of airborne LiDAR relies on road design and topographic conditions. However, the application is considered minimal. This airborne LiDAR application is believed to have potential to be implemented for Malaysian roads featuring a highland topography, steep slopes, and forested areas. Figure 3 and Table 4 shows the overall future direction of the use of airborne LiDAR technology in civil engineering road design.

**Table 4.** Future direction of airborne LiDAR technology in civil engineering road design.

| Future Direction | Author(s) | Year | LiDAR Data | Application/Software/Modelling | Findings |
|---|---|---|---|---|---|
| Minimise road environmental impact | A.E. Akay et al. | 2014 | DTM | 3D Forest Road Alignment Optimization | To calculate quantity of sediment from road systems. |
| | Contreras et al. | 2012 | DEM | The computerized model | To reduce the volume of cutting at the slope. |
| | J.M Vilbig et al. | 2014 | DEM | The quasi-network flow | To reduce the volume of cutting at the slope. |
| | W. Chen et al. | 2014 | DTM | LiDAR data and slope images | Assist in identifying a landslide-prone region. |
| | T. Görüm | 2019 | DTM | Landslide Inventory | Reduces landslides in a roadwork area. |
| | M. Saito et al. | 2013 | DTM | Shallow Landslide Risk Map | To consider shallow slides near the road. |
| Designs roads in Slope and Hilly Areas | P. Jagodnik et al. | 2020 | DTM | Soil Geological Mapping | To detect gravitational mass movement and rockfall occurrence. |
| | R. Pellicani et al. D. Gadone et al. | 2019 2018 | DTM | Mapping | To identify characteristics that signal soil failure and detect soil instability. |
| | C. Chen et al. | 2018 | DEM | Progressive TIN Densification (PTD) | To help detect ground movement conditions |
| Locating road drainage structures | J.B Lindsay and K. Dhun | 2015 | DEM | Least-Cost Breaching | To determine surface drainage channel patterns, particularly those found under bridges and culverts. |
| | R. Li et al. | 2012 | DEM | LiDAR-derived hydrologic | To determine drainage patterns. |
| | J. Roelans et al. | 2018 | DEM | LiDAR Dropout Modelling | To determine drainage ditches. |
| | J.B Lindsay et al. | 2019 | DEM | Feature-Preserving Smoothing Method | To determine drainage pattern flow. |

**Table 4.** *Cont.*

| Future Direction | Author(s) | Year | LiDAR Data | Application/Software/Modelling | Findings |
|---|---|---|---|---|---|
| Maintaining road connectivity | M. Sharma et al. | 2020 | DSM, BEM | Terrain parameter extraction | For road mapping. |
| | S. Buján et al. | 2021 | DTM | Road-mapping | To aid in road connection and new road design. |
| | L. Barazzetti et al. | 2020 | DTM | Model of road networking | To connect existing roads and new roads. |
| | K. Karila et al. | 2016 | DTM | LiDAR-based road-mapping | For road network monitoring. |
| Topographic data for road planning and design | B. Matinnia et al. | 2017 | DEM | LiDAR road mapping | To aid in road design without time-consuming field surveying methods in broad areas. |
| | B.P Reis et al. | 2019 | DTM, DSM, nDSM | LiDAR road networking | Produce the main images in places with dense tree cover to sustain forest regeneration and road network. |
| | B. Bigdeli et al. | 2018 | DTM | Invasive Weed Optimization | To improve accuracy when creating DTM at the forested canopy and steep terrain. |
| | N. Fareed and C. Wang | 2021 | DEM | Drainage Structure Mapping Algorithm | To obtain culvert modified DEMs for hydrological studies on highways. |
| | R. Pu and S. Landry | 2020 | nDSM | Multi-seasonal Pléiades satellite imagery and LiDAR data | To provide accurate tree mapping for maintaining sustainable development area. |

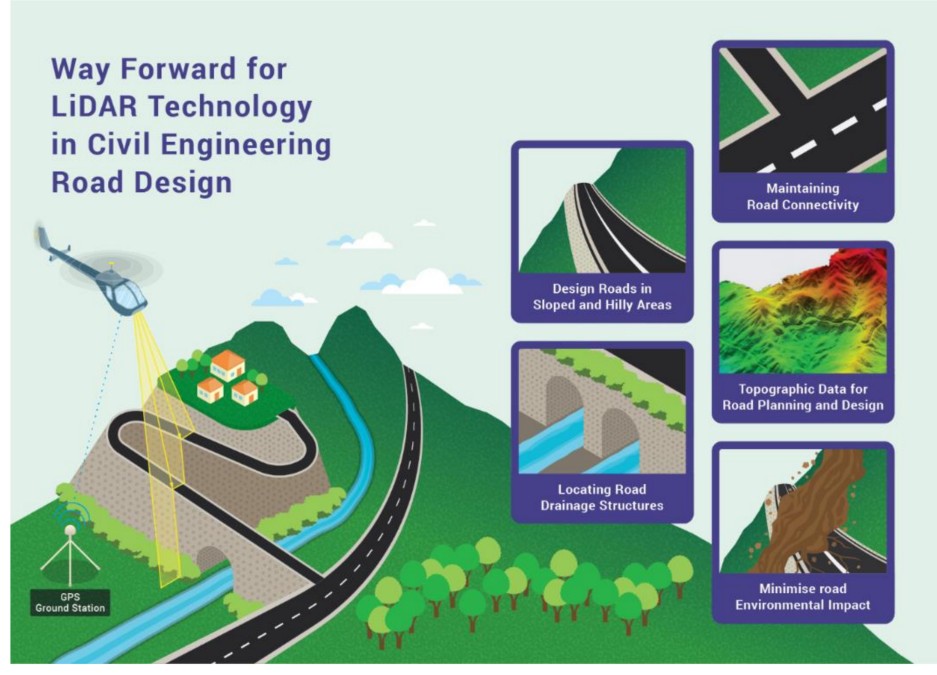

**Figure 3.** Future direction of LiDAR technology in civil engineering road design.

**7. LiDAR Technology for Roadway Inspection and As-Built Documentation**

Along with the rapid development of technology, LiDAR is seen to be increasingly used for roads and transportation. Its advantages have been proven and can be used to measure road projects. Then, close monitoring regarding the design can be carried out after the project is completed. Several studies have been conducted, as listed below, that further discuss this matter.

*7.1. Inventory Road Asset*

LiDAR can be used as a benchmark in the field of road asset management [101]. In the study by S.A. Gargoum and K. El Basyouny [102,103], a LiDAR point extraction procedure managed to detect road signs and was used as an inventory for highway assets. Then, He et al. [104] used an extraction from LiDAR points to provide up-to-date data for transportation networks, particularly highway assets (traffic sign, light pole, bridge, culvert, and billboard). N. Fareed and C. Wang [94] in their study used the Drainage Structure Mapping algorithm to obtain information about the location of drainage structures such as bridges and culverts. Meanwhile, M.Javanmardi et al. [105] utilized histogram height information along a road to detect traffic signs by looking into coloured images representing traffic signs.

*7.2. Road Inspection*

In road areas with mountainous terrain, data from LIDAR are essential to detect the features of mass movement due to the fact that it provides early indications in areas with a risk of landslides, as stated in the study by J.U.H Eitel et al. [106]. Meanwhile, P. Lo et al. [107] used LiDAR data as a geological survey to detect tunnels and slope failure threats in road network locations with a tunnel structure that spanned the underground rock mass area. C.H. Yeh et al. [108] found that LiDAR data aided in providing dip–slope mapping in highway areas, addressing the problem of the information gained by visual interpretation in dip–slope locations being inaccurate.

In the studies by R. Chen et al., [109], S. Chhatkuli et al. [110] and P. Caudal et al. [111], information derived from LiDAR data assisted in the detection of deep-seated gravitational slope deformation, which is standard for hilly routes and poses a concern to highway users in the event of landslides or rockslides. Meanwhile, C. Lee et al. [112] found that LiDAR can map existing locations as well as areas that may collapse in the future, such as constricted valley areas, tension cracks, or a landslide foot, especially on mountain roads.

*7.3. Road Geometry Assessment*

In the study by C. Wen [113], LiDAR data assessment was able to aid in road detection while collecting information about existing roads, which is extremely valuable in road geometry assessment and road network planning. Meanwhile, Y. Ma et al. [114] stated that the visual condition of drivers in high-risk areas could be inspected using an approach which combines the Modified Delaunay Triangulation algorithm and a Back-Propagation neural network to estimate the sight distance of LiDAR data. Furthermore, S. Ural et al. [115] mentioned that the extraction of LiDAR data has aided transportation agencies in meeting their requirement for precise road geometric information that is regularly updated for future road design planning.

*7.4. Road Modelling*

For the formation of road modelling, the study by A. Ferraz et al. [5] employed Profile Analysis and Object-Based Image Analysis to create a 3D geometric characterization representing each section and road for road modelling.

*7.5. Importance of LiDAR Technology for Roadway Inspection and As-Built Documentation*

From the above studies, the advantages of airborne LiDAR in the monitoring of constructed road can be looked into upon the completion of road construction, which

aids in preparing as-built drawings. LiDAR technology simplifies the process of checking and evaluating that a road's geometry, after being built by transportation agencies, complies with road design criteria and specification. LiDAR technology also assists in the mapping of road furniture and structured assets position on the road. Overall, it is beneficial to designers and practitioners in the planning, design, and construction of road projects. Figure 4 and Table 5 show LiDAR technology for roadway inspection and as-built documentation.

**Table 5.** LiDAR Technology for roadway inspection and as-built documentation.

| Road work | Author(s) | Year | LiDAR Data | Findings |
|---|---|---|---|---|
| Inventory road asset | S.A. Gargoum and K. El Basyouny | 2019 2019 | Point cloud | LiDAR point extraction procedure to detect road signs that can be used as an inventory for highway assets. |
| | He et al. | 2017 | Point cloud | Extraction from LiDAR points to provide up-to-date data for transportation networks, particularly highway assets. |
| | N. Fareed and C.Wang | 2021 | DEM Point cloud | The Drainage Structure Mapping algorithm is used to acquire data on the location of drainage structures. |
| | M. Javanmardi et al. | 2021 | Airborne georeferenced colour images Noisy data | Histogram height information to detect traffic signs. |
| Road inspection | J.U.H Eitel et al. | 2016 | DEM | LIDAR data to detect the characteristic features of mass movement. |
| | P. Lo et al. | 2021 | DEM | LiDAR data as a geological survey to detect tunnels and slope failure. |
| | C.H. Yeh et al. | 2017 | DEM | LiDAR data provide dip–slope mapping. |
| | R. Chen et al., S. Chhatkuli et al. P. Caudal et al. | 2015 2016 2016 | DEM | LiDAR data can aid with the detection of deep-seated gravitational slope deformation. |
| Road geometry assessment | C. Lee et al. | 2017 | DEM DTM | LiDAR data to map existing locations as well as areas that may collapse in the future. |
| | C. Wen | 2020 | DTM | LiDAR data can aid in road detection. |
| | Y. Ma et al. | 2019 | DSM DTM | Modified Delaunay Triangulation algorithm and a Back-Propagation neural network to estimate sight distance in LiDAR data. |
| | S. Ural et al. | 2015 | DEM DSM nDSM | The extraction of LiDAR to aid in precise road geometric information and future road design planning. |
| Road modeling | BA.Ferraz et al. | 2015 | DTM | Profile analysis and Object-Based Image Analysis to create 3D geometric characterization representing sections and roads width information for road modelling. |

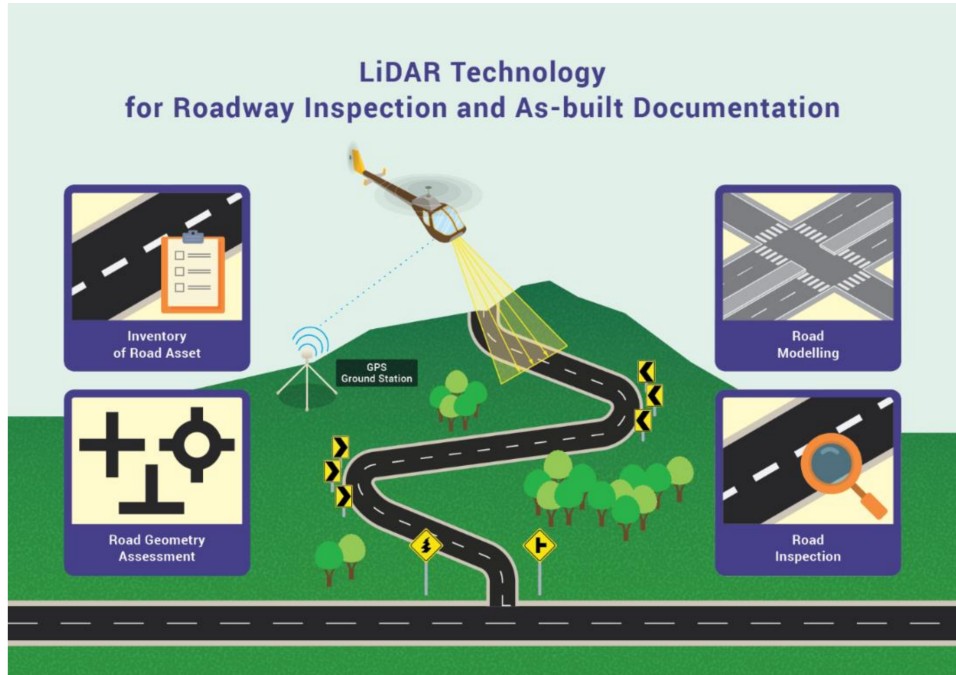

**Figure 4.** LiDAR technology for roadway inspection and as-built documentation.

## 8. Limitations

Despite the study's contributions to the perceived efficiency of LiDAR technology in road design and management, it has several limitations. User satisfaction with LiDAR technology for road designers is one of these limitations. Even though a road designer may have a civil engineering background, they may still be unfamiliar with the LiDAR technology used in surveying. However, they may expert in the interpretation of conventional topographic data by using TSs. Thus, it is essential to conduct research to address this issue to examine whether it is beneficial to both the engineer and the surveyor.

Another limitation is the risk to pilots, specifically when aircraft and helicopters pass by difficult areas [82]. UAV technology can be used to overcome this problem and avoid risks posed to the pilot, co-pilot, and technician. UAV technology is equipped with an RGB camera integrated with a multispectral sensor and LiDAR sensor. It is able to give accurate readings [116–118]. The combination of UAV and LiDAR provides excellent data [21]. Data from LiDARs are able to produce earthworks information [119], whereas images from UAVs provide a clear picture of the study area [120,121]. Apart from that, the issue of limited budget allocation is raised in the small and medium-scale designs for using airborne LiDAR [19].The use of UAVs is considered as another good alternative for this situation [122].

Lastly, LiDAR technology can generate complete data to produce as-built drawings, but not in automatic way. BIM-GIS integration is achieved when the road information is extracted from LiDAR data and thus integrated with BIM through BIM parameterization in the GIS layer where road information is made into a BIM object [90].

## 9. Conclusions

The application of airborne LiDAR is significant in road design management due to the fact that it consists of complete data information which influences road planning. Aside from road planning, it also creates a comprehensive road networking network for systematic road design. Road design management becomes more manageable when it is possible to track the location of roads and new roads. As such, the location of drainage structures can be precisely detected. Indeed, the accurate terrain data information gained

from LiDAR reduces the risk of road construction in areas which are prone to landslides or rockfalls.

In addition, road management becomes much better when LiDAR information helps with road inspection and the design of roads which are ready for construction. It is suggested that government agencies should propose road inventory, road geometry assessment, and road modelling to be used as complete data for government transportation agencies without site locations.

However, despite the effectiveness of this technology, further research is still required on the use of this technology in Malaysia. There is still no specific study on road design using airborne LiDAR being conducted in Malaysia compared to other countries. Thus, the importance of further research works that address the lack of acceptance of this technology is highlighted. Based on the technology acceptance factors that previous researchers have studied, it can be concluded that there is a need to increase the usage of this technology by both government agencies and the private sector. We urge top management to continue providing specific disclosures regarding the effectiveness of technology as the technology provides accurate information to complete work in an easier way. Furthermore, proper rules are required to ensure that LiDAR work is standardized and to create a standard to be followed in order to create high-quality topographic data for road planning and transportation.

**Author Contributions:** Conceptualization, F.H.A., M.A.K., K.N.A.M. and A.A.; methodology, F.H.A., M.A.K., K.N.A.M. and A.A.; validation, F.H.A., M.A.K., K.N.A.M. and A.A.; formal analysis, F.H.A., M.A.K., K.N.A.M. and A.A.; writing—original draft preparation, F.H.A.; writing—review and editing, F.H.A., M.A.K., K.N.A.M. and A.A.; visualization, F.H.A., M.A.K., K.N.A.M.; supervision, M.A.K., K.N.A.M. and A.A.; project administration, F.H.A., M.A.K., K.N.A.M. All authors have read and agreed to the published version of the manuscript.

**Funding:** This research was funded by Dana Impak Perdana with grant no. DIP-2018-030.

**Institutional Review Board Statement:** Not applicable.

**Informed Consent Statement:** Not applicable.

**Data Availability Statement:** The data sets during and/or analysed during the current study available from the corresponding author on reasonable request.

**Acknowledgments:** The author would like to thank Universiti Kebangsaan Malaysia (UKM) for the financial support under Dana Impak Perdana (DIP-2018-030).

**Conflicts of Interest:** The authors declare no conflict of interest.

## Abbreviations

The following abbreviations are used in this manuscript:

| | |
|---|---|
| DEM | Digital Elevation Model |
| DSM | Digital Surface Model |
| DTM | Digital Terrain Model |
| nDTM | Normalized Digital Terrain Model |
| BEM | Bare Earth Model |
| LiDAR | Light Detection and Ranging |
| UAV | Unmanned Aerial Vehicle |
| TS | Total Station |
| RGB | Red–Green–Blue |

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
