# Peer review of "Perceived Usefulness of Airborne LiDAR Technology in Road Design and Management: A Review"

_sustainability, doi:10.3390/su132111773_

Round 1
Reviewer 1 Report
The following points can be considered for further enhancement of the paper.
- Authors have claimed that “Perceived Usefulness of Airborne LiDAR Technology in Road Design and Management: A Review”. It is nowhere mentioned about the name of road design and management used. How Usefulness of Airborne LiDAR Technology used , what are new challenges ? Many statements are not transparent nor adequately justified.
- The RELATED WORKS is paced at the end of the manuscript. It should be kept as Section II. This section should be presented exhaustively while incorporating the critical relevant and related research papers. May key papers are not cited in this manuscript. kindly add recently work.
- figure 1 did not well explain. also discuss the each entity of figure.
- Author did not well analyzed the limitations and previous proposed methods benefits.
- Author should add possible solutions existing methods .
- lack of discussion and add future work section.
- Many grammatical errors are present in the paper.
Author Response
Hello, sir/madam. Thank you for your thoughtful feedback and suggestion. It is very beneficial to me. Please see the attachment.

Reviewer 2 Report
The paper reviews variables and items from the perceived usefulness of LiDAR technology in road design and management.
This is an appreciable effort, however, what seems to be missing is the accuracy of various applications of LiDAR in civil engineering road design, as well as a comparative assessment with similar technologies in terms of accuracy, cost, and overall performance.
Despite the fact that the authors refer to certain limitations of LiDAR technology, to what extent can weather, topography, plantation and urban-rural settings affect the accuracy of the results? Can LiDAR technology be combined with other contemporary technologies?
Author Response

(The authors gave the same response as above.)

Reviewer 3 Report
sustainability-1371856
The manuscript surveys application of the airborne LiDAR to civil engineering problems, in particular in road design and management. There is an extensive list of references with more than 120 items. However, paper is “dry”. There is no enough useful information about respective techniques and results. Papers are mainly compared by claims without any in-depth analysis with the classification of the techniques and methods. It requires more details to be useful for researchers, Ph.D. students, engineers working on respective projects. OK
The paper organization also is not particularly good. Conclusions from other papers are given word by word in the text and repeated in tables in the same manner without any critical assessment.
Author Response

(The authors gave the same response as above.)

Round 2
Reviewer 1 Report
All comments are well addresses.
Author Response
Thank you very much Prof./Dr./ Sir/Madam for your prior comments. It helps a lot.
Please see the attachment .

Reviewer 2 Report
my comments have been successfully addressed.
Author Response
Thank you very much Prof./Dr./ Sir/Madam for your prior comments.
Reviewer 3 Report
Revision of
sustainability-1371856
The manuscript passed through a set of extensive modifications. It is now acceptable for publication. Probably due to a number of changes there are many typographical and other mistakes that should be corrected: two dots in the abstract, missing dot between [25] and The LiDAR, abbreviation DSM is introduced twice, once mistakenly instead for “elevation model” (see Section 3.0), “ompletion time” in Section 5.3, extra dot in the table on page 18, missing spaces on many places in front of references, Section 5.1 title has a typo, missing dot after [120] in Section 8.0, etc. Concentration on issues of application of the LiDAR in the native country is a good point of the revised paper.
Author Response
Thank you very much Prof/ Dr./Sir/ Madam for your prior comments. I've improved lots. Please see the attachment
